# The effects of Parkinson's disease, music training, and dance training on beat perception and production abilities

Prisca Hsu[1]*, Emily A. Ready[2,3], Jessica A. Grahn[2,3]

**1** Schulich School of Medicine and Dentistry, Western University, London, Ontario, Canada, **2** Department of Psychology, Western University, London, Ontario, Canada, **3** Brain and Mind Institute, Western University, London, Ontario, Canada

* phsu26@uwo.ca

## Abstract

Humans naturally perceive and move to a musical beat, entraining body movements to auditory rhythms through clapping, tapping, and dancing. Yet the accuracy of this seemingly effortless behavior varies widely across individuals. Beat perception and production abilities can be improved by experience, such as music and dance training, and impaired by progressive neurological changes, such as in Parkinson's disease. In this study, we assessed the effects of music and dance experience on beat processing in young and older adults, as well as individuals with early-stage Parkinson's disease. We used the Beat Alignment Test (BAT) to assess beat perception and production in a convenience sample of 458 participants (278 healthy young adults, 139 healthy older adults, and 41 people with early-stage Parkinson's disease), with varying levels of music and dance training. In general, we found that participants with over three years of music training had more accurate beat perception than those with less training (p < .001). Interestingly, Parkinson's disease patients with music training had beat production abilities comparable to healthy adults while Parkinson's disease patients with minimal to no music training performed significantly worse. No effects were found in healthy adults for dance training, and too few Parkinson's disease patients had dance training to reliably assess its effects. The finding that musically trained Parkinson's disease patients performed similarly to healthy adults during a beat production task, while untrained patients did not, suggests music training may preserve certain rhythmic motor timing abilities in early-stage Parkinson's disease.

## Introduction

Most humans naturally perceive the underlying temporal regularity in music termed the beat. Humans often spontaneously synchronize their body movements to music through tapping or clapping. The process of synchronizing, or entraining, movement to the beat engages motor areas of the brain. In particular, the basal ganglia have been shown to play a key role in perceiving the beat [1], and Parkinson's disease patients, who have dysfunctional inputs to the basal

**Data Availability Statement:** The data underlying the results presented in the study are available from figshare: 10.6084/m9.figshare.19209702.

**Funding:** The project is supported by the Natural Science and Engineering Research Council of

Canada (JAG) (RGPIN-2016-05834) and the James S. McDonnell Foundation (JAG). The funders had no role in study design, data collection and analysis, decision to publish, or preparation of the manuscript.

**Competing interests:** The authors have declared that no competing interests exist.

ganglia, show specific beat perception impairments [2]. Beyond Parkinson's disease, however, even neurotypical individuals show a striking range in how accurately they both perceive and synchronize to a beat. Some of this variability is related to past experiences, such as music and dance training [3, 4]. It is therefore possible that perception or production deficits in Parkinson's disease may be offset by a music or dance background.

Neural timing networks involve both cortical and subcortical motor control areas [5, 6]. These cortical structures include the premotor cortex and supplementary motor area (SMA), and the subcortical structures include the basal ganglia and the cerebellum [5]. The basal ganglia are affected in Parkinson's disease, a neurodegenerative disease characterized by progressive cell death of dopaminergic neurons in the substantia nigra, resulting in loss of excitatory stimulation of a part of the basal ganglia called the putamen [7]. The disruption of dopamine projection within these networks appears to result in beat processing deficits [2, 8, 9]. Patients are impaired on tapping tasks involving finger tapping to a metronome followed by paced tapping without a metronome [8] and more complex rhythm discrimination tasks that required participants to decipher whether two beat-based rhythms were identical [2]. These deficits may be related to dopamine levels in the basal ganglia, as Parkinson's disease patients improve on rhythm discrimination tasks after taking dopaminergic medication [9]. Similar trends of impaired temporal discrimination performance are observed among healthy adults when dopamine uptake is disrupted by dopamine receptor antagonists [10]. Furthermore, beat perception and production deficits are correlated with idiopathic REM sleep disorder which commonly occurs prior to Parkinson's disease onset and is often considered a prodromal-Parkinson's disease symptom [11]. These studies confirm the crucial role of dopamine in timing and rhythm processing.

## Music training

The effects of music training have been studied previously by comparing musicians and non-musicians. Musicians can distinguish changes in beat-based and nonbeat-based rhythms better than non-musicians [12]. Similarly, musicians are better able to extract metrical structures from music [13]. Aside from perceptual advantages, music training relates to better motor and timing abilities. On a simple tapping task, musicians demonstrated lower tapping variability and more accurate synchronization to the external rhythmic stimulus [14]. In addition, musicians were more sensitive to tempo changes in the stimulus and exhibited faster phase correction compared to non-musicians [14]. Recently developed behavioral batteries of rhythm tasks such as the Battery for the Assessment of Auditory Sensorimotor and Timing Abilities (BAASTA) and the Harvard Beat Assessment Test (H-BAT) corroborate other work showing that music training is associated with better beat processing abilities [15, 16].

## Dance training

Similar to musicians, dancers also have superior timing perception abilities compared to non-dancers [17]. However, unlike musicians, dancers are especially skilled at entraining their movements to visual events. Dancers must synchronize their movements to the music and with the movements of other dancers, resulting in elevated motor entrainment abilities with both auditory and visual stimuli [18]. Dancers learn choreography by watching others perform and later fine-tuning their movements to match with other dancers [19]. Dancers engage both visual and motor networks during their training and are found to be better at extracting a beat from visual stimuli compared to musicians [20]. In addition, viewing dance movements enhances auditory meter perception in dancers suggesting visual-auditory entrainment abilities [21]. Dancers are experts in whole-body coordination to auditory cues. A study comparing

dancer and non-dancer muscle contractions at the onset of salient metronome beats found that dancers had more accurate movements compared to non-dancers [22]. Overall, dancers have exceptional whole-body sensorimotor entrainment and motor coordination, both dynamic properties that are thought to contribute to their superior visual-motor and auditory-motor entrainment abilities [23].

## Study rationale

Though beat perception and production are thought to be related, there is evidence that the abilities dissociate [24, 25], thus many assessments include both perceptual and production tasks [16, 26, 27]. The Beat Alignment Test (BAT) is one such assessment [26]. The beat perception task of the BAT has been extensively used in various studies [15, 24] and has successfully identified people with impaired beat perception but intact beat production, and vice versa [24, 25]. It is simple and brief, therefore especially useful in testing clinical populations in which complex or fatiguing tasks are less feasible. Recent research has used the BAT to assess sensorimotor integration as well as beat perception and production in Parkinson's disease patients [9, 28].

Beat perception and production accuracy are positively impacted by music and dance training but negatively impacted by neurological changes in Parkinson's disease. Previous studies have used the BAT to assess rhythmic abilities in healthy and clinical groups. We expect training to relate to better rhythmic abilities in healthy adults, but whether this advantage is preserved in Parkinson's disease is unknown. In this study, the BAT was used to measure beat perception and production in the three participant groups: healthy young adults, healthy older adults, and people with early-stage Parkinson's disease, all groups with varying levels of music and dance training. The data were aggregated from several studies conducted over a period of several years, in which all individuals completed the BAT and a demographics questionnaire in the context of other studies. We hypothesized that music and dance training would correlate with better beat perception and production abilities, while Parkinson's disease would reduce these abilities relative to controls. The behavioral benefits of music and dance training may still be preserved in Parkinson's disease. Therefore, Parkinson's disease patients with previous training may have better abilities than patients without training.

## Methods

### Participants

278 healthy young adults (M = 20.41, SD = 3.01), 139 healthy older adults (M = 64.63, SD = 9.27) and 41 people with early-stage Parkinson's disease (M = 68.28, SD = 7.73) were recruited for various music and gait studies conducted in the Music and Neuroscience Lab. People with early-stage Parkinson's Disease (Hoehn & Yahr stages 2–3) were recruited from the community of Southwestern Ontario through community outreach and flyers. Given the exploratory nature of the study, Parkinson's disease patients were not excluded based on medication regimen, years since diagnosis, or having received deep brain stimulation. Six participants who did not complete both beat perception and production tasks of the BAT and eight participants who did not indicate the years of previous music and dance training experience were excluded from the analyses. The final N = 458 only includes participants who completed the BAT and indicated the years of previous music and dance training experience. Participants of each group varied in level of music and dance training (Table 1). Based on years of training, participants were divided into two categories: 0–2 years and 3+ years. This threshold was chosen to balance sample sizes across participants with different degrees of music and dance training while maintaining the distinction between minimal and more extensive training. Informed

**Table 1. Participant demographics.**

| | N = 458 | Age | Music training (years) | | Dance training (years) | |
|---|---|---|---|---|---|---|
| | | Years (SD) | 0–2 | 3+ | 0–2 | 3+ |
| Young adults | 278 | 20.41 (3.01) | 111 | 167 | 208 | 70 |
| Older adults | 139 | 64.63 (9.27) | 71 | 68 | 111 | 28 |
| Parkinson's disease patients | 41 | 68.28 (7.73) | 25 | 16 | 40 | 1 |

consent was obtained from all participants and approval for all studies was obtained from the Western Medical Research Ethics Board or the Western Nonmedical Research Ethics Board (104487,106385).

## Stimuli

Musical stimuli were taken from the Beat Alignment Test of the Goldsmiths Musical Sophistication Index (Gold-MSI) v1.0 [26] downloaded from https://www.gold.ac.uk/music-mind-brain/gold-msi/download/. Version 1.0 of the Gold-MSI is optimized relative to the version reported in Müllensiefen et al., (2014) [26], with 17 items, selected as described in documentation available at https://www.gold.ac.uk/music-mind-brain/gold-msi/download/ (S1 Fig). Musical excerpts were chosen from a variety of music genres and ranged from 10 to 16 seconds in duration. In the beat perception task, beeps were superimposed on the music excerpts 5 seconds into the music. The BAT was administered on a PC laptop using E-Prime (2.0) software (Psychology Software Tools, 2002). Auditory stimuli were delivered through Sennheiser HD 280 headphones. All participants completed both beat perception and production tasks in one session.

## Beat perception task

Participants listened to musical excerpts (3 practice trials, 17 test trials) with superimposed metronome beeps either on or off the beat. Off-beat excerpts could either result from beeps coming in too early or too late relative to the actual beat (phase error), or from beeps too fast or too slow relative to the tempo of the actual beats (period error). Phase shifts of the superimposed beeps were adjusted 10% or 17.5% ahead relative to the musical beat. Period shifts of the superimposed beeps were adjusted 2% slower or faster relative to the musical tempo (S1 Fig). Participants were tasked to identify whether the superimposed beeps were "on the beat" or not, without using body movement to assist the judgement. The trial order was randomized, and participants rated how confident they were of their judgment after each excerpt on a 7-point Likert scale.

## Beat production task

Participants heard the same musical excerpts as the beat perception task with the superimposed beeps removed. Each participant synchronized their finger tapping as soon as they perceived the beat of the music. Each excerpt was presented twice consecutively. The order of the musical excerpts was randomized, and participants were asked to rate their familiarity with each musical excerpt on a 7-point Likert scale. The extent to which participants matched their tapping to the actual beats was measured based on phase and tempo accuracy, as well as tapping variability.

Phase matching accuracy was represented by asynchrony (Eq 1), which measured the absolute difference between tap time and nearest beat position. The asynchrony score was obtained by taking the mean of the absolute difference between each tap and its nearest beat divided by

the mean inter-beat interval (IBI). IBI was calculated by subtracting consecutive beat onsets. High asynchrony scores reflect high tapping phase error, indicating that participants tapped too early or too late relative to the actual beat. In contrast, low asynchrony scores reflected taps that were more aligned with the musical beat. Asynchrony scores were averaged across the 17 trials to obtain an average asynchrony score for each participant.

$$asynchrony = \frac{mean_{|response-beat|}}{mean_{IBI}} \tag{1}$$

Tempo matching accuracy was represented by the coefficient of deviation (CDEV) score (Eq 2), which measured the absolute deviation between inter-response interval (IRI) and inter-beat interval (IBI). The inter-response interval was determined by subtracting consecutive tap onset times. High CDEV scores reflect high tapping period error, meaning that participants either tapped too fast or too slow relative to the actual beat tempo. In contrast, low CDEV scores reflected more accurate tempo matching abilities. CDEV scores were averaged across the 17 trials to obtain an average CDEV score for each participant.

$$CDEV = \frac{mean_{|IRI-IRB|}}{mean_{IBI}} \tag{2}$$

Tapping variability was represented by the coefficient of variation (CoV) score (Eq 3), which measured motor response variability independent of the stimuli. High CoV scores reflect less consistent tapping, whereas low CoV scores reflect more consistently paced tapping. CoV scores were averaged across the 17 trials to obtain an average CoV score for each participant.

$$CoV = \frac{SD_{|IRI|}}{mean_{IRI}} \tag{3}$$

## Demographic questionnaire

After the beat perception and production tasks, participants filled out a demographic questionnaire to describe their age and years of music and dance training experience. This exploratory study employed convenience sampling of participants that partook in different music and walking studies. Consequently, healthy young and older adults were given a general demographic questionnaire while Parkinson's disease patients were given the Gold-MSI Questionnaire [26].

## Statistical analyses

Beat perception ability was quantified by the percent of correct responses. Beat production was quantified by asynchrony and coefficient of deviation scores reflecting phase matching and tempo matching accuracy, respectively, as well as coefficient of variation scores reflecting tapping variability. To investigate the effects of music training, 3 (healthy young adults, healthy older adults, Parkinson's disease patients) x 2 (0–2 years, 3+ years of music training) ANOVAs were conducted on beat perception and production measures. We excluded the Parkinson's disease group from the dance training analysis because only one Parkinson's disease patient reported 3+ years of dance experience. For this reason, a separate 2 (healthy young adults, healthy older adults) x 2 (0–2 years, 3+ years of dance training) x 2 (0–2 years, 3+ years of music training) ANOVA was conducted to investigate the combined effects of dance training and music training on beat perception and production in the healthy adult groups, excluding

the Parkinson's disease group. For this ANOVA, only main effects of or interactions with dance are reported, as music training effects are covered in the 3 x 2 ANOVA that enabled the inclusion of the Parkinson's disease group. Follow-up ANCOVAs using age, music training and participant groups as covariates were conducted to investigate the dance training effects. Main effects and interactions were confirmed by follow-up simple main effect and post-hoc pairwise comparisons using Bonferroni correction. In addition to traditional frequentist statistical approaches, we conducted two Bayesian ANOVAs for each dependent variable similar to the traditional ANOVAs. Bayes hypothesis testing allows for the distinction between "absence of evidence" (i.e., the data is not informative, design underpowered) or "evidence of absence" (i.e., the data supports the null hypothesis), allowing for a more informed understanding of the results. One Bayesian ANOVA included music training levels and all three groups, and the other included dance training levels and only the young/older adult groups, to further quantify support for the null versus experimental hypotheses. A 2 x 2 x 2 Bayesian ANOVA with music training, dance training, and young/older groups produced similar outcomes to the individual music and dance training 2x2 Bayesian ANOVAs, so is not reported. Data were analyzed and visualized using JASP and R software.

## Results

The mean age for older adults (64.5) significantly differed from that of Parkinson's disease patients (68.2), as shown by a Welch's unequal variance two-sample t-test ($t = -2.58$, $p = .012$). However, linear models fitting age and beat perception, asynchrony, coefficient of variation and coefficient of deviation indicated age did not predict performance on any of the dependent variables in these groups. Therefore, age differences in the range found in the older adults and Parkinson's disease group do not appear to reliably affect beat perception or production (all p's > .05).

### Beat alignment test perception scores

Participants with 0–2 years and 3+ years of music training averaged 61% (SD = 0.16) and 70% (SD = 0.17) correct responses, respectively. The 3 (group) x 2 (music training) ANOVA revealed a main effect of music training [$F(1, 457) = 20.42$, $p < .001$, $\eta p^2 = 0.043$]. Participants with greater music training, regardless of participant group, demonstrated more accurate beat perception compared to those with minimal music training. The main effect of music training was qualified by a Music training x Group interaction [$F(1, 457) = 3.49$, $p = .031$, $\eta p^2 = 0.015$]. Simple main effects revealed that for both young adults and Parkinson's patients, those with more extensive music training differed from those with minimal music training [young adults: $F(1, 277) = 30.06$, $p < .001$, $\eta p^2 = 0.098$; Parkinson's: $F(1,40) = 7.62$, $p = .006$, $\eta p^2 = 0.16$], but there was no reliable effect of music training for older adults [$F(1, 138) = 1.04$, $p = .31$, $\eta p^2 = 0.0075$] (Fig 1A). No main effect of or interactions with dance training were noted (all p's>.05) in the 2 (healthy adult groups) x 2 (music training) x 2 (dance training) ANOVA (Fig 2A).

### Beat alignment test production scores

**Beat production: Phase matching.** The 3 (group) x 2 (music training) ANOVA revealed a significant main effect of music training [$F(1, 457) = 8.86$, $p = .003$, $\eta p^2 = 0.019$] and a Music training x Group interaction [$F(1, 457) = 4.19$, $p = .016$, $\eta p^2 = 0.018$]. Follow-up simple main effects indicated that Parkinson's patients with minimal music training had lower phase matching accuracy than Parkinson's patients with more extensive training [$F(1, 41) = 10.07$, $p = .002$, $\eta p^2 = 0.20$]. Parkinson's patients with minimal training also had lower phase

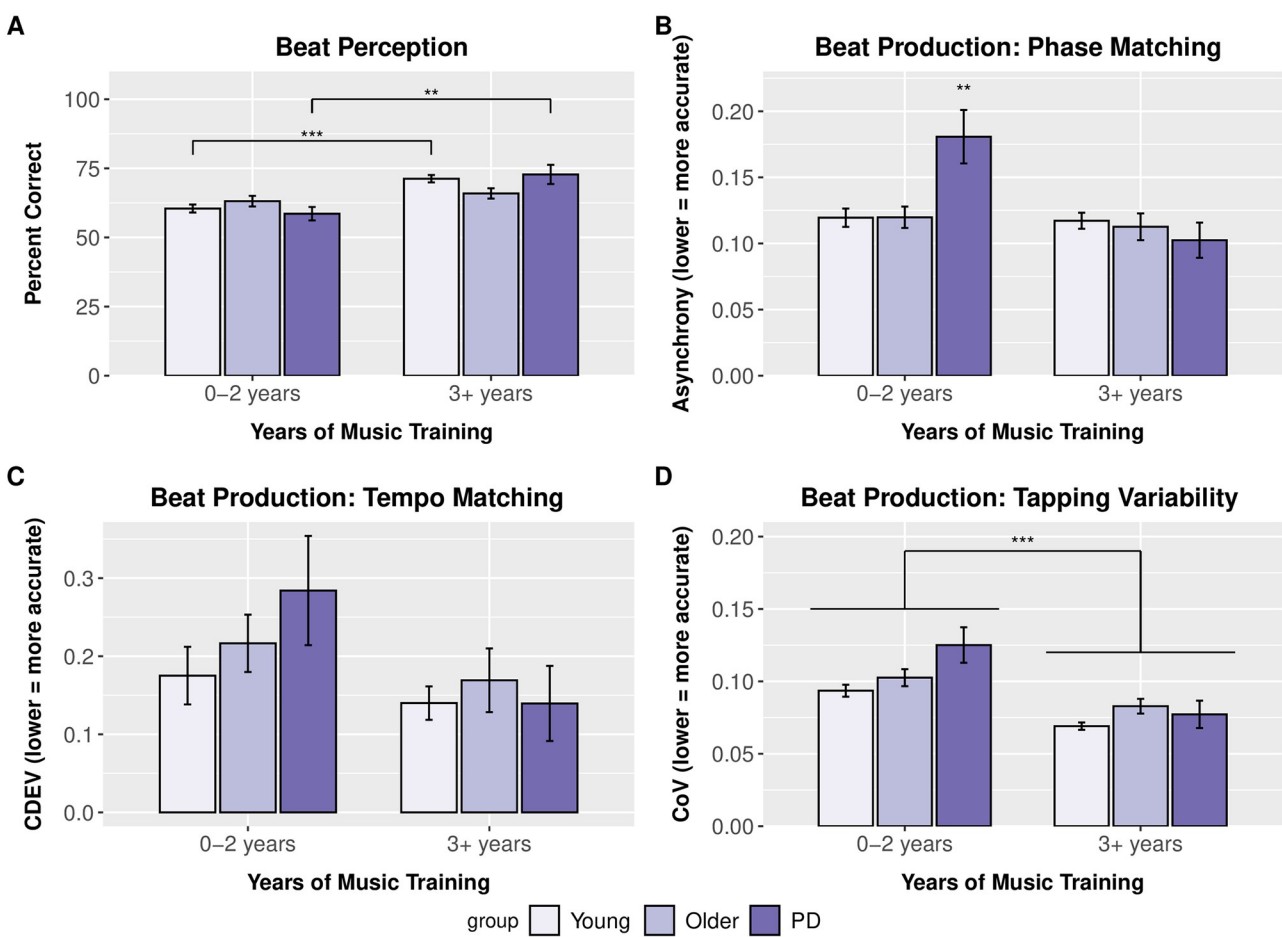

**Fig 1. Music training effects on beat perception and production.** Performance broken down by group (young, older, Parkinson's disease) and music training (0–2 years, 3+ years) for beat perception (A), beat production phase matching (B), beat production tempo matching (C), and beat production tapping variability (D). For beat perception, young adults and Parkinson's disease patients with more extensive music training were significantly better than those without. For asynchrony (phase matching), Parkinson's disease patients with minimal music training were significantly worse than all other groups. No significant differences were present for coefficient of deviation (tempo matching). For coefficient of variation (tapping variability), older adults and Parkinson's patients were more variable than younger adults, and participants with more extensive music training (regardless of group) were less variable than those with little training. Error bars indicate the standard error of the mean. ** = $p < .01$, *** = $p < .001$.

matching accuracy than young adults and older adults in both music training groups (all p-values < .05) (Fig 1B). Interestingly, Parkinson's patients with music training performed similarly to healthy adults with training, suggesting that music training was associated with retained beat production abilities for Parkinson's disease patients. The 2 (healthy adult groups) x 2 (music training) x 2 (dance training) ANOVA revealed no significant dance training effects or interactions in healthy young and older adults (Fig 2B).

**Beat production: Tempo matching.** There were no significant effects in the 3 (group) x 2 (music training) ANOVA (Fig 1C). The 2 (healthy adult groups) x 2 (music training) x 2 (dance training) ANOVA revealed no main effects of or interactions with dance training (all p's > .05) (Fig 2C).

**Beat production: Tapping variability.** The 3 (group) x 2 (music training) ANOVA revealed a main effect of music training [$F(1,457) = 33.28$, $p < .001$, $\eta p^2 = 0.069$] and a main effect of group [$F(1, 457) = 6.00$, $p = 003$, $\eta p^2 = 0.026$] (Fig 1D). The 2 (healthy adult groups) x

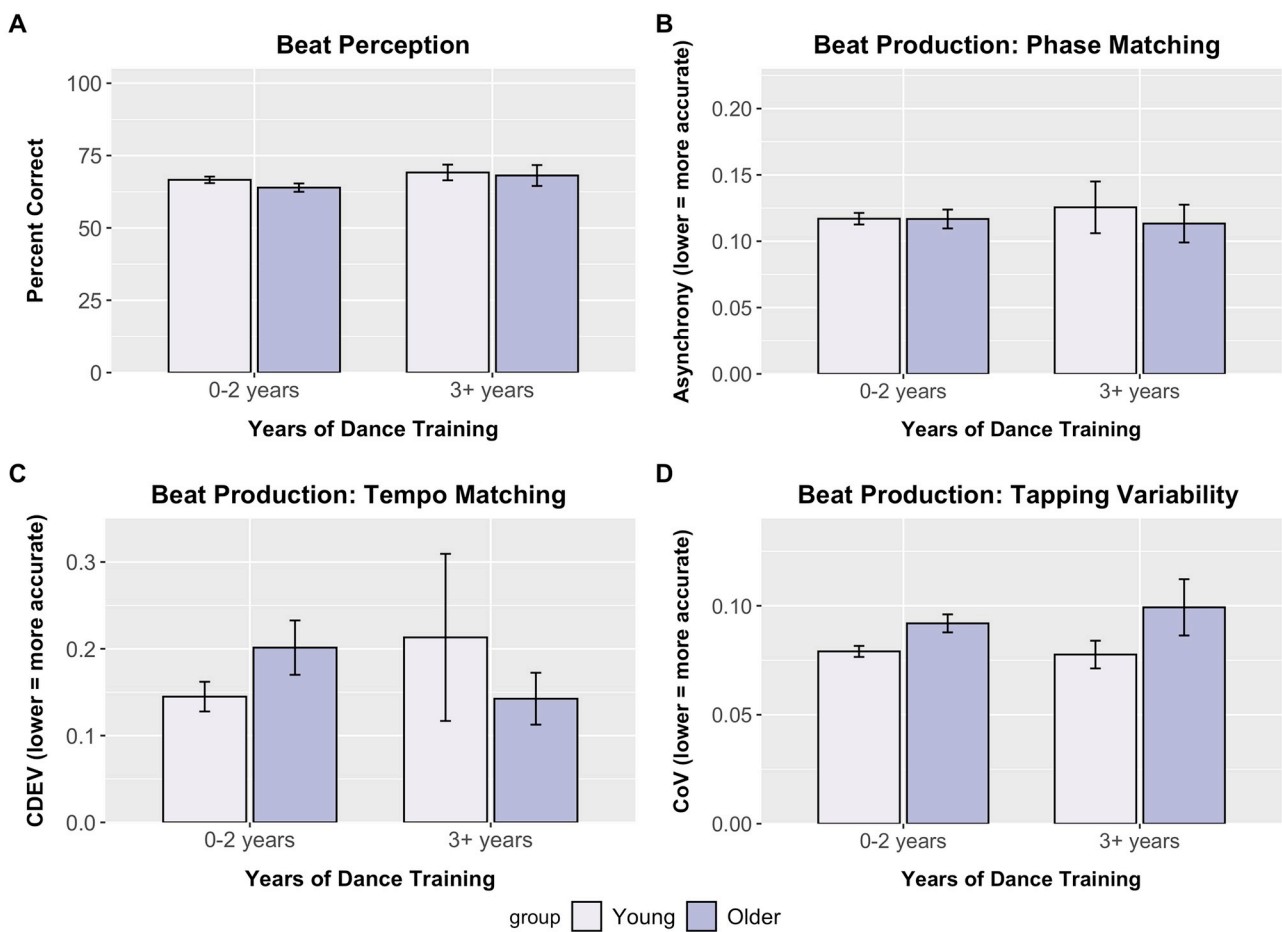

**Fig 2. Dance training effects on beat perception and production.** Performance broken down by group (young, older, Parkinson's disease) and dance training (0–2 years, 3+ years) for beat perception (A), beat production phase matching (B), beat production tempo matching (C), and beat production tapping variability (D). No significant differences were present for beat perception, phase matching (asynchrony) and tempo matching (coefficient of deviation). Tapping variability did differ between groups. Error bars indicate the standard error of the mean.

2 (music training) x 2 (dance training) ANOVA revealed no significant dance training effects or interactions in healthy young and older adults (Fig 2D).

## Bayesian analyses

The Bayesian ANOVA compares the predictive performance of each model with and without each independent variable and their interactions. The P(M) column is the prior model probability which assumes that all rival models are equally likely to represent the data. The P(M|data) column indicates the probability of each model given the actual data. The $BF_m$ column indicates the relative likelihood of each model compared to the average of all other models, and the $BF_{10}$ column indicates the relative likelihood of each model compared to the null model. As $BF_{10}$ deviates from 1, support for the null or alternative hypothesis increases. Generally, $0.33 < BF_{10} < 3$ indicates that the data is insufficient to support either null or alternative hypothesis. $0.1 < BF_{10} < 0.33$ provides moderate support for the null hypothesis and $3 < BF_{10} < 10$ provides moderate support for the alternative hypothesis. Finally, a $BF_{10} < 0.1$ provides strong support for the null hypothesis and $BF_{10} > 10$ provides strong support for the

Table 2. Comparison of Bayes models: Music & group.

| Models | P(M) | P(M\|data) | BF $_M$ | BF $_{10}$ | error % |
|---|---|---|---|---|---|
| Null model | 0.20 | 0.00 | 0.00 | 1.00 | |
| music | 0.20 | 0.89 | 33.23 | 685845 | 0.00 |
| music + group + music*group | 0.20 | 0.06 | 0.26 | 46220 | 1.85 |
| music + group | 0.20 | 0.05 | 0.20 | 36339 | 1.48 |
| group | 0.20 | 0.00 | 0.00 | 0.12 | 0.02 |

Note: P(M) = prior model probability; P(M|data) = posterior model probability; BF$_m$ = change from prior to posterior model odds; BF$_{10}$ = Bayes Factor in favor of each model compared with the null model. Music = music training level, group = young/older/Parkinson's group.

Table 3. Comparison of Bayes models: Dance & group.

| Models | P(M) | P(M\|data) | BF $_M$ | BF $_{10}$ | error % |
|---|---|---|---|---|---|
| Null model | 0.20 | 0.57 | 0.57 | 1.00 | |
| dance | 0.20 | 0.19 | 0.19 | 0.33 | 0.00 |
| group | 0.20 | 0.17 | 0.17 | 0.29 | 0.00 |
| dance + group | 0.20 | 0.06 | 0.06 | 0.10 | 1.08 |
| dance + group + dance*group | 0.20 | 0.01 | 0.01 | 0.02 | 2.54 |

Note: P(M) = prior model probability; P(M|data) = posterior model probability; BF$_m$ = change from prior to posterior model odds; BF$_{10}$ = Bayes Factor in favor of each model compared with the null model. Dance = dance training level, group = young/older group.

alternative hypothesis [29, 30]. Strength of the evidence can be quantified based on the Bayes Factor (e.g., BF$_{10}$ = 8 is twice as strong as BF$_{10}$ = 4 in supporting the alternative hypothesis).

**Beat perception.** When assessing music training and group differences on beat perception, the model including only the music training factor was more supported than the null model by a Bayes Factor of 685845, which is strong evidence (Table 2). Likewise, models including music and group main effects, as well as music and group main effects plus the music*group interaction had large BF$_{10}$ values of 46220 and 36339, respectively (Table 2).

When assessing the effects of dance training on beat perception, the dance model had a BF$_{10}$ value of 0.33 (Table 3), indicating insufficient support for either the null hypothesis or the alternative hypothesis [29, 30].

**Beat production: Phase matching.** For music training, music and group models revealed a BF$_{10}$ value between 0.33 and 3 (Table 4), suggesting that the data was not well represented by any of the provided models [29]. While the traditional 3 x 2 ANOVA above revealed a music*group interaction, the effect size was small, driven by Parkinson's disease non-musicians differing from Parkinson's disease musicians and healthy adults. Thus, both analyses

Table 4. Comparison of Bayes models: Music & group.

| Models | P(M) | P(M\|data) | BF $_M$ | BF $_{10}$ | error % |
|---|---|---|---|---|---|
| Null model | 0.20 | 0.40 | 2.72 | 1.00 | |
| music | 0.20 | 0.25 | 1.30 | 0.61 | 0.03 |
| music + group + music*group | 0.20 | 0.15 | 0.73 | 0.38 | 0.00 |
| music + group | 0.20 | 0.12 | 0.55 | 0.30 | 1.80 |
| group | 0.20 | 0.08 | 0.33 | 0.19 | 8.34 |

**Table 5. Comparison of Bayes models: Dance & group.**

| Models | P(M) | P(M\|data) | BF $_M$ | BF $_{10}$ | error % |
|---|---|---|---|---|---|
| Null model | 0.20 | 0.76 | 12.86 | 1.00 | |
| dance | 0.20 | 0.13 | 0.59 | 0.17 | 0.00 |
| group | 0.20 | 0.09 | 0.40 | 0.12 | 0.00 |
| dance + group | 0.20 | 0.02 | 0.06 | 0.02 | 2.15 |
| dance + group + dance*group | 0.20 | 0.00 | 0.02 | 0.01 | 5.91 |

**Table 6. Comparison of Bayes models: Music & group.**

| Models | P(M) | P(M\|data) | BF $_M$ | BF $_{10}$ | error % |
|---|---|---|---|---|---|
| Null model | 0.20 | 0.60 | 5.97 | 1.00 | |
| music | 0.20 | 0.30 | 1.72 | 0.50 | 0.00 |
| music + group + music*group | 0.20 | 0.07 | 0.31 | 0.12 | 0.02 |
| music + group | 0.20 | 0.03 | 0.11 | 0.04 | 2.69 |
| group | 0.20 | 0.00 | 0.01 | 0.00 | 2.09 |

indicate that the Music training x Group interaction was small, and, in the Bayesian analysis, insufficient to power the selection of music training and group as the best overall model. However, the Bayesian analysis provides additional information that the null hypothesis was also not strongly supported by the data.

For dance training effects on phase matching, the dance model had a BF$_{10}$ of 0.17 (Table 5), indicating moderate support for the null hypothesis.

**Beat production: Tempo matching.** For tempo matching, the music model had a BF$_{10}$ of 0.50 (Table 6), indicating insufficient support for either the null hypothesis or alternative hypothesis. For dance, BF$_{10}$ = 0.22 (Table 7), indicating moderate support for the null hypothesis.

*Beat production*: *Tapping variability*. When assessing music training and group differences on tapping variability, the music and group model was more supported than the null model by a Bayes Factor of 5.30e+9, followed by the full model with interaction (music and group plus music*group interaction) with a Bayes Factor of 1.19e+9 and the music model with a Bayes Factor of 2.65e+8 (Table 8). The music and group model has the largest Bayes Factor and thus suggests that it is most likely to accurately reflect the variance seen in the data. When assessing dance training and group differences on tapping variability, the group and dance model was more supported than the null model by a Bayes Factor of 2.63, which is weak evidence. However, unlike other measures of beat perception, the group model was best supported with a Bayes Factor of 16.78 (Table 9).

**Table 7. Comparison of Bayes models: Dance & group.**

| Models | P(M) | P(M\|data) | BF $_M$ | BF $_{10}$ | error % |
|---|---|---|---|---|---|
| Null model | 0.20 | 0.69 | 8.73 | 1.00 | |
| dance | 0.20 | 0.15 | 0.71 | 0.22 | 0.00 |
| group | 0.20 | 0.12 | 0.56 | 0.18 | 0.00 |
| dance + group | 0.20 | 0.03 | 0.11 | 0.04 | 1.12 |
| dance + group + dance*group | 0.20 | 0.01 | 0.05 | 0.02 | 1.44 |

**Table 8. Comparison of Bayes models: Music & group.**

| Models | P(M) | P(M\|data) | BF $_M$ | BF $_{10}$ | error % |
|---|---|---|---|---|---|
| Null model | 0.20 | 0.00 | 0.00 | 1.00 | |
| music + group | 0.20 | 0.79 | 14.90 | 5.30e+9 | 7.12 |
| music + group + music*group | 0.20 | 0.17 | 0.84 | 1.19e+9 | 1.75 |
| music | 0.20 | 0.04 | 0.16 | 2.65e+8 | 0.00 |
| group | 0.20 | 0.00 | 0.00 | 301.27 | 0.02 |

## Discussion

This study examined the effects of music and dance training on beat perception and production abilities across the life span and in the context of Parkinson's disease. We predicted that music and dance training would improve beat perception and production skills, while the neurological deficits associated with Parkinson's disease would negatively affect these skills. We further predicted that some positive impacts of music and dance training would be preserved despite disease state. Indeed, on the beat perception task, young adults and Parkinson's disease patients who were musically trained did better than those who weren't, but this difference was not significant for older adults. For beat production, only the Parkinson's groups differed as a function of musical training: Parkinson's patients with music training performed comparably to healthy adults, whereas patients with minimal training showed significantly worse phase matching accuracy (high asynchrony scores). However, tempo matching accuracy was not affected by music training in any group. Furthermore, no dance training effects were found, although the dance training analysis was restricted to older and younger adults, as only one Parkinson's patient had 3+ years of dance training. Finally, interpretation of the reported null results from the traditional frequentist statistical analyses is aided by Bayesian analyses indicating that true null results are unlikely, suggesting that greater power may be necessary to detect effects of dance training or music.

We observed that music training was associated with better beat perception in young adults and Parkinson's disease patients, but not older adults. While it is widely documented in the literature that music training is associated with better beat perception, the results obtained in the current study may suggest that music training effects on beat perception could decay over time since training, as evidenced by the significant differences seen in healthy young adults, but not healthy older adults. However, this decay either does not apply to the Parkinson's patients, or other factors are at play, as music training was associated with better performance for patients.

Contrary to our hypothesis, Parkinson's disease patients were not significantly impaired on the beat perception task. These results were consistent with Cameron and colleagues' findings that beat perception tested on the BAT did not differ across Parkinson's disease and healthy control groups [9]. In contrast, our results differed from Benoit et al.'s findings that

**Table 9. Comparison of Bayes models: Dance & group.**

| Models | P(M) | P(M\|data) | BF $_M$ | BF $_{10}$ | error % |
|---|---|---|---|---|---|
| Null model | 0.20 | 0.05 | 0.20 | 1.00 | |
| group | 0.20 | 0.79 | 15.02 | 16.78 | 0.00 |
| dance | 0.20 | 0.12 | 0.56 | 2.62 | 1.68 |
| dance + group | 0.20 | 0.03 | 0.13 | 0.68 | 1.37 |
| dance + group + dance*group | 0.20 | 0.01 | 0.03 | 0.16 | 0.00 |

Parkinson's patients showed worse timing perception than healthy adults [31]. However, in their study, Parkinson's patients were tasked to detect misaligned beats in a two-measure music excerpt, as opposed to several seconds of tones overlaid on music in the BAT. This task is less taxing on memory and attention, and thus may not be able to differentiate between healthy adult groups and Parkinson's patients. Rhythm perception was also examined in Cameron et al.'s study using a rhythm discrimination task (not part of the BAT). In their work, the rhythm discrimination task was more sensitive to timing perception deficits in Parkinson's disease than BAT. While both BAT and rhythm discrimination tasks measure beat perception abilities, there are important differences. Beat perception in the discrimination task relies solely on temporal information, without the additional beat cues afforded by music in the BAT (e.g., pitch, harmony, timbral, and amplitude cues). The rhythm discrimination task also involves working memory to compare consecutively presented rhythms. In contrast, the BAT relies on a comparison between simultaneous temporal sequences (musical stimuli and overlaid tones), and it does not require attending to the stimuli for the entire duration nor remembering them. Cochen de Cock et al. corroborated these findings by suggesting that cognitive abilities such as attention, executive function and cognitive flexibility could influence beat perception abilities [32]. Thus, the mechanisms required to perform the beat perception task in the BAT could be intact in Parkinson's disease while mechanisms for strictly temporally-based rhythm perception and comparison could be impaired.

We predicted that rhythm-intensive training, such as through dance or music, would improve beat perception skills. Surprisingly, only music training elicited positive effects on beat perception. Although dancers generally performed better on the beat perception test than non-dancers, the differences were not significant, consistent with other studies that tested young adults on the BAT [33]. Importantly, it appears unlikely that years of training differed between musicians and dancers. The Gold-MSI groups music training into multi-year levels, and the average level for the musicians was 6–9 years. The dance questionnaire requested specific numbers, and the 3+ years group averaged 8.1 years. Thus, the lack of dance training effect relative to musical training effect seems unlikely to be caused by differences in years of training. Of course, years of training is an imprecise quantification of true training effects, as the rigor of different training programs and hours of deliberate practice varies across individuals.

We found that Parkinson's disease patients performed worse than healthy adults on the beat production task, however, this appeared to be influenced by music training. Parkinson's disease patients with minimal music training exhibited lower phase matching accuracy than healthy adults, as reflected by their increased asynchrony scores. However, decreased accuracy on the motor tasks overall may reflect nothing more than the generalized effects of Parkinson's disease, consistent with previous findings that patients tap more variably than healthy controls [34]. Interestingly, Parkinson's disease patients with more extensive music training exhibited phase matching similar to that of healthy adults, unlike patients with minimal training. This suggests that music training may have influenced motor control or temporal accuracy. Though the literature on music training effects in the Parkinson's disease population is limited, these results were consistent with the trends seen in healthy adult populations [35, 36]. Older adults in our study (regardless of whether they have Parkinson's disease) displayed higher tapping variability compared to younger adults, and musicians (regardless of group) displayed more consistent tapping. These results were consistent with trends seen in Thompson et al.'s cross-sectional study [37]. However, an additional music*group interaction is plausible provided that the model representing music and group and their interaction is well supported by the 3x2 Bayesian ANOVA. Contrary to previous findings that Parkinson's disease patients tapped

either faster or slower on synchronization and self-paced timing tasks [38], patients in this study did not demonstrate significantly different tapping tempos compared to healthy adults.

People with minimal (0–2 years) dance training did not differ from those with more extensive (3+ years) dance training on beat production. We performed another analysis using a stricter cutoff for dance training (0–5 vs. 6+ years) but still did not find any significant differences. These results contrast previous findings on rhythm entrainment which found that dancers were better at synchronizing whole-body movements to a recurring beat compared to non-dancers [23]. However, the beat production task may contain components that are emphasized more by music training than dance training. For example, dancers often use visual cues, observing other dancers, to fine-tune their entrainment [19]. Furthermore, dancers use more whole-body movements when synchronizing with music rather than finger-tapping [14, 22]. Future studies could investigate both auditory-motor and visual-motor entrainment to better understand the effects of dance training on motor entrainment skills.

Parkinson's disease-related motor symptoms are most commonly treated using pharmaceutical therapies, such as levodopa, MAO-B inhibitors and dopamine agonists [39]. Medications help manage some motor symptoms but don't necessarily improve gait symptoms, such as shuffling and freezing [40]. Therefore, Parkinson's disease patients are frequently treated with rehabilitative therapies; rhythmic auditory stimulation (RAS) is a common therapy aimed at improving gait patterns. RAS provides temporal cues, such as a metronome, to which a person can entrain their walking pattern. Better rhythm processing abilities correlate with better RAS outcomes in both healthy and Parkinson's disease populations [32, 41]. Therefore, Parkinson's disease patients with music training may be better candidates for music- and rhythm-based therapies. The current results suggest it may be possible to adapt the BAT to be an effective screening for Parkinson's disease patients who might benefit from rhythm-based interventions by identifying those with intact beat processing abilities.

The current study employed a convenience sample from multiple studies. Parkinson's disease patients completed the Gold-MSI questionnaire, which groups music training into categories, (e.g., 0, 1, 2, 3, 4–5, 6–9 years) whereas healthy adults completed a survey that simply asked for the number of years in which they engaged in regular music practice. Music training in healthy adults was grouped to match Parkinson's disease patients to analyze the entire dataset. To minimize sample size disparity, music training categories were grouped once more to create two reasonably sized groups (0–2 vs. 3+ years) while maintaining the distinction between minimal and more extensive training. Few participants reported dance training experience, so two sizable groups (0–2 vs. 3+ years) were created for statistical analysis purposes while excluding Parkinson's disease patients from this analysis due to low sample size. In addition, most Parkinson's disease participants were receiving dopaminergic therapy, which improves beat perception and production abilities [9, 38]. Therefore, greater group differences may be found in an off-medication paradigm. Hoehn and Yahr stage is also the only indicator of disease severity collected in the study, and future work addressing the impact of these factors across disease stages would benefit from including more specific disease severity information such as disease duration or the levodopa equivalent daily dose and by including participants with moderate to severe stages of Parkinson's.

## Conclusion

Our findings indicate significantly better beat perception and production skills among participants with more extensive music training. Parkinson's disease patients with more extensive music training exhibited better beat perception and production skills than patients with minimal training. These results contribute to the growing knowledge of the long-term effects of

music training and suggest that music training may preserve certain motor timing functions related to beat processing in early-stage Parkinson's disease.

## Supporting information

**S1 Fig. Stimuli used in beat alignment perception test v1.0.** Note: The order of the musical excerpts was randomized.
(PDF)

## Acknowledgments

The authors would like to thank Brittany S. Roberts for helping with data collection. We would also like to thank the healthy adults and the people with Parkinson's disease as well as their families for their participation.

## Author Contributions

**Conceptualization:** Prisca Hsu, Emily A. Ready, Jessica A. Grahn.

**Data curation:** Prisca Hsu, Emily A. Ready.

**Formal analysis:** Prisca Hsu.

**Funding acquisition:** Jessica A. Grahn.

**Investigation:** Prisca Hsu, Emily A. Ready.

**Methodology:** Prisca Hsu, Emily A. Ready, Jessica A. Grahn.

**Project administration:** Prisca Hsu.

**Resources:** Emily A. Ready, Jessica A. Grahn.

**Supervision:** Emily A. Ready, Jessica A. Grahn.

**Validation:** Emily A. Ready, Jessica A. Grahn.

**Visualization:** Prisca Hsu.

**Writing – original draft:** Prisca Hsu.

**Writing – review & editing:** Prisca Hsu, Emily A. Ready, Jessica A. Grahn.

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
