## [Decision Letter · Decision Letter 0]

23 Nov 2021

PONE-D-21-33421The effects of Parkinson’s disease, music training, and dance training on beat perception and production abilitiesPLOS ONE

Dear Dr. Hsu,

Thank you for submitting your manuscript to PLOS ONE. After careful consideration, we feel that it has merit but does not fully meet PLOS ONE’s publication criteria as it currently stands. Therefore, we invite you to submit a revised version of the manuscript that addresses the points raised during the review process.

We look forward to receiving your revised manuscript.

Kind regards,

Sonja Kotz

Academic Editor

PLOS ONE

Journal Requirements:

Reviewers' comments:

Reviewer's Responses to Questions

**Comments to the Author**

1. Is the manuscript technically sound, and do the data support the conclusions?

Reviewer #1: Yes

Reviewer #2: Yes

2. Has the statistical analysis been performed appropriately and rigorously? 

Reviewer #1: Yes

Reviewer #2: Yes

3. Have the authors made all data underlying the findings in their manuscript fully available?

Reviewer #1: Yes

Reviewer #2: Yes

4. Is the manuscript presented in an intelligible fashion and written in standard English?

Reviewer #1: Yes

Reviewer #2: Yes

5. Review Comments to the Author

Reviewer #1: Thank you for giving me the opportunity to review this article. This study provides an insightful view of the role of music and dance training on the change in rhythmic skills with age. The authors tested rhythm perception and tapping synchronization with musical material in three different groups (healthy adults, healthy older adults, and older adults with Parkinson’s disease) with or without music or dance training. The main finding is that Parkinsonian participants with music training performed as well as healthy adults in synchronization task.

The article is well written, and the methodology is appropriate. My main concern is that the authors did not use an index of tapping variability per se (e.g., coefficient of variation of the asynchronies). They claim that the asynchrony score reflects variability and consistency, which is not exact. I suppose that additional group differences would be highlighted by tapping variability, and this might change the overall findings pattern. Please see below for additional suggestions about this point.

Here are the specific comments that, I think, should be addressed by the authors to improve the manuscript:

Introduction

- “simple timing tasks [8] and more complex beat-based rhythm discrimination tasks [2].” (l65). What are the simple tasks that the authors refer to? Interval-based? It would be helpful to describe briefly the tasks typically used in these experiments here. In the same paragraph, two other types of tasks are introduced (“rhythm discrimination tasks”, l67, and “temporal discrimination task”, l69). Please clarify by giving a short description of the tasks, as very few readers will be familiar with that.

- Note that in some studies, the BAT is only the perceptual task ([11], [13], [16] in the article reference list), whereas in others it includes a variety of other tasks ([14] in the article reference list). It would be worth mentioning it to avoid possible confusion.

- The second paragraph of the “Rhythm disturbance in Parkinson’s disease” section does not seem to fit in well with the section. The authors can consider moving it to another subsection.

- Because the article does not include any brain imaging measures, I suggest not to start by describing the brain areas involved in PD, music training, and dance training in the different sections. Focusing on the behavior is more relevant. Neuronal underpinnings can be briefly mentioned at the end of each paragraph, probably in one sentence, especially because they are not discussed in the rest of the article.

- “As the basal ganglia have not been implicated in the neural changes associated with music and dance training” (l126). I would tone down this statement. It may be true that no changes have been directly observed in the basal ganglia, and I trust the authors who have certainly carefully reviewed the literature on this topic. Nevertheless, this absence of observed change in the basal ganglia may have different causes, and the basal ganglia are part of networks that are associated with music and dance training.

Methods

- Is there a significant difference in age between the older adults and the PD groups?

- “Six participants who did not complete both beat perception and production tasks of the BAT and participants who did not indicate the years of previous music and dance training experience were excluded from the analyses” (l140-142). Does the final N of participants presented in table 1 include those participants? If so, I suggest to rather present only the participants that were included in the analyses.

- The initial battery developed by Müllensiefen, Gingras, Musil, & Stewart (2014) that is used in this study contains 18 stimuli. Why did the authors use only 17? I would find it useful to give some precision about the stimuli (e.g., N of trials with phase and period changes, percentage of change etc.) so that the reader does not have to read the article by Müllensiefen et al. to understand the methods. This is particularly important if the number of stimuli differs.

- “low asynchrony scores reflected less variable and more consistent tap times” (l180). I do not think the asynchrony score reflect variability and consistency. It tells how far from the beat the participants tapped; a high score can reflect a very consistent performance in antiphase, for example. None of the two indexes reflect variability; it could be worth using a coefficient of variation of the asynchronies [SD(asynch)/Mean(asynch)], where asynch = Mean(Response - Beat). This index might reveal additional effects that the tempo matching index failed to capture. Notably, I expect that participants with dance and music training are less variable in their tapping performance. For example, a recent study on children with cerebellar anomalies showed that a dance training protocol reduced their variability in a synchronization task (Bégel et al., 2021).

Results

- “When assessing the effects of dance training on beat perception, the null model had a BF10 of 1 (Table 2b), suggesting that the data could not discriminate between the alternative hypothesis from the null hypothesis” (l288-290). This is also the case for the music & group model. Please explain a bit more how you come to this conclusion and what is the difference between the two Null models. This remark applies to the phase matching analyses as well.

- Figure 2. Because the PD group was not included in the analyses with dance training, I think their results should not appear in the figure.

- Figure 3. It is not clear what the three lines (user, wide, ultrawide) are. Most readers are not familiar with Bayesian statistics. It is important to give more information.

Discussion

- The absence of difference between the PD and healthy adults groups in beat perception is inconsistent with the results of Benoit et al. (2014). There are interindividual difference in Parkinson’s patients that may explain the contradictory results (Cochen de Cock et al., 2018). I think this should be mentioned in the discussion.

- The “Implications” and “Limitations” sections are quite long. Condensing them into one section seems appropriate.

References

Bégel, V., Bachrach, A., Dalla Bella, S., Laroche, J., Delval, A., Riquet, A., & Dellacherie, D. (2021). Dance improves motor, cognitive and social skills in children with developmental cerebellar anomalies. The Cerebellum, Advance Online Publication

Benoit, C. E., Dalla Bella, S., Farrugia, N., Obrig, H., Mainka, S., & Kotz, S. A. (2014). Musically cued gait-training improves both perceptual and motor timing in Parkinson’s disease. Frontiers in Human Neuroscience, 8, 494.

De Cock, V. C., Dotov, D. G., Ihalainen, P., Bégel, V., Galtier, F., Lebrun, C., ... & Dalla Bella, S. (2018). Rhythmic abilities and musical training in Parkinson’s disease: do they help?. NPJ Parkinson's disease, 4(1), 1-8.

Reviewer #2: Hsu et al. explored in a large population, the effects of music and dance training on beat perception and production abilities across the life span and in the context of Parkinson’s disease, predicting that music and dance training would improve beat perception and production skills, while Parkinson’s disease would negatively affect these skills.

They observed that young adults and Parkinson’s disease patients who were musically trained did better than those who weren’t in beat perception tasks. This effect was not significant in older adults.

For beat production, only Parkinson’s patients with music training performed comparably to healthy adults, whereas patients with minimal training had significantly higher asynchrony scores.

Tempo matching was not modified by musical training and dance training surprisingly had no effect.

The results of the study are surprising since the expected effects of musical training are limited, and the dance training were not observed.

The main limitation is that the information about the training is very limited: we don’t know when the training was and what its intensity and level was. This an important limitation in this study that is discussed in the discussion

There is also an important lack of information on Parkinson’s disease. Hoehn and Yahr stage is very insufficient to evaluate PD severity. Disease duration Ldopa Equlivalent daily dose are needed to understand the kind of patients that were explored. These two information should be added.

Also the recent reference on rhtythm disturabances in PD and prePD should be added and discussed

Rhythm disturbances as a potential early marker of Parkinson's disease in idiopathic REM sleep behavior disorder.

Cochen De Cock V, de Verbizier D, Picot MC, Damm L, Abril B, Galtier F, Driss V, Lebrun C, Pageot N, Giordano A, Gonzalvez C, Homeyer P, Carlander B, Castelnovo G, Geny C, Bardy B, Dalla Bella S.Ann Clin Transl Neurol. 2020 Mar;7(3):280-287. doi: 10.1002/acn3.50982. Epub 2020 Feb 14.PMID: 32059086

6. PLOS authors have the option to publish the peer review history of their article (what does this mean?). If published, this will include your full peer review and any attached files.

Reviewer #1: **Yes: **Valentin Bégel

Reviewer #2: No

---

## [Author Response · Author response to Decision Letter 0]

6 Jan 2022

Dear Dr. Sonja Kotz,

Thank you for handling our manuscript, “The effects of Parkinson’s disease, music training, and dance training on beat perception and production abilities,” for possible publication in Plos One. We also thank the reviewers for their careful feedback and helpful comments. We now submit a revised version, with additions about beat production tapping variability. Point-by-point responses to the reviewers’ comments are attached below. Note that the line numbers refer to the Revised Manuscript with Tracked Changes file. We hope that you might find this version to be suitable for publication for the wide readership of Plos One.

Sincerely,

Prisca Hsu, Emily Ready, Jessica Grahn

Reviewer #1

Introduction

- “simple timing tasks [8] and more complex beat-based rhythm discrimination tasks [2].” (l65). What are the simple tasks that the authors refer to? Interval-based? It would be helpful to describe briefly the tasks typically used in these experiments here. In the same paragraph, two other types of tasks are introduced (“rhythm discrimination tasks”, l67, and “temporal discrimination task”, l69). Please clarify by giving a short description of the tasks, as very few readers will be familiar with that.

Indeed, there is room to clarify what is meant by simple and complex here. We have added task clarification in lines 64-66: 

Patients are impaired on tapping tasks involving finger tapping to a metronome followed by paced tapping without a metronome [8] and more complex rhythm discrimination tasks that required participants to decipher whether two beat-based rhythms were identical [2]. 

- Note that in some studies, the BAT is only the perceptual task ([11], [13], [16] in the article reference list), whereas in others it includes a variety of other tasks ([14] in the article reference list). It would be worth mentioning it to avoid possible confusion.

Yes, good point. Reference 11,13 and 16 in the original submission, only used the perception component of the BAT and used another, different, synchronization task to measure beat production. We have now noted that these studies only utilized the beat perception component of the BAT. 

L137-142: Though beat perception and production are thought to be related, there is evidence that the abilities dissociate [24,25], thus many assessments include both perceptual and production tasks [16,26,27]. The Beat Alignment Test (BAT) is one such assessment [26]. The beat perception task of the BAT has been extensively used in various studies [15,24] and has successfully identified people with impaired beat perception but intact beat production, and vice versa [24,25].

- The second paragraph of the “Rhythm disturbance in Parkinson’s disease” section does not seem to fit in well with the section. The authors can consider moving it to another subsection.

We agree that the second paragraph seems more about the relationship between production/perception and the BAT, not on rhythm disturbances in PD and not about the neural underpinnings. We have moved this paragraph to the Study Rationale section.

- Because the article does not include any brain imaging measures, I suggest not to start by describing the brain areas involved in PD, music training, and dance training in the different sections. Focusing on the behavior is more relevant. Neuronal underpinnings can be briefly mentioned at the end of each paragraph, probably in one sentence, especially because they are not discussed in the rest of the article.

We agree that the descriptions of anatomical differences between musicians/non-musicians and dancers/non-dancers aren’t relevant to the current study, we have now removed that information. 

- “As the basal ganglia have not been implicated in the neural changes associated with music and dance training” (l126). I would tone down this statement. It may be true that no changes have been directly observed in the basal ganglia, and I trust the authors who have certainly carefully reviewed the literature on this topic. Nevertheless, this absence of observed change in the basal ganglia may have different causes, and the basal ganglia are part of networks that are associated with music and dance training.

Fair point. We have tempered the sentence accordingly:

L155-159: We hypothesized that music and dance training would correlate with better beat perception and production abilities, while Parkinson’s disease would reduce these abilities relative to controls. The behavioral benefits of music and dance training may still be preserved in Parkinson’s disease. Therefore, Parkinson’s disease patients with previous training may have better abilities than patients without training.

Methods

- Is there a significant difference in age between the older adults and the PD groups?

The mean age for older adults is 64.5 and Parkinson’s disease patients is 68.2. A Welch two-sample t-test does show a significant difference between mean ages for the two groups (p = .01), but linear models fitting age and beat perception, asynchrony, coefficient of variation and coefficient of deviation indicate no significant effects of age on the dependent variables (all p’s > .05). We believe that the age difference between older adults and Parkinson’s disease patients is therefore not driving any group differences in beat perception or production. 

Each graph is showing age of the participant along the x-axis and the relevant dependent variable along the y-axis. These graphs include data points from both healthy older and Parkinson’s groups.

We have added this information to our results section, clarifying that there is a significant difference between the ages of healthy older adults and Parkinson’s patients, but this age difference does not reliably affect in beat perception and production abilities. 

L298-317: The mean age for older adults (64.5) significantly differed from that of Parkinson’s disease patients (68.2), as shown by a Welch’s unequal variance two sample t-test (t = -2.58, p = .012). However, linear models fitting age and beat perception, asynchrony, coefficient of variation and coefficient of deviation indicated age did not predict performance on any of the dependent variables in these groups. Therefore, age differences in the range found in the older adults and Parkinson’s disease group do not appear to reliably affect beat perception or production (all p’s > .05).

- “Six participants who did not complete both beat perception and production tasks of the BAT and participants who did not indicate the years of previous music and dance training experience were excluded from the analyses” (l140-142). Does the final N of participants presented in table 1 include those participants? If so, I suggest to rather present only the participants that were included in the analyses.

The final N=458 does not include participants who did not complete both beat perception and production tasks or did not indicate years of music/dance training. We have clarified this in the Participants section.

L169-173: Six participants who did not complete both beat perception and production tasks of the BAT and eight participants who did not indicate the years of previous music and dance training experience were excluded from the analyses. The final N=458 only includes participants who completed the BAT and indicated the years of previous music and dance training experience.

- The initial battery developed by Müllensiefen, Gingras, Musil, & Stewart (2014) that is used in this study contains 18 stimuli. Why did the authors use only 17? I would find it useful to give some precision about the stimuli (e.g., N of trials with phase and period changes, percentage of change etc.) so that the reader does not have to read the article by Müllensiefen et al. to understand the methods. This is particularly important if the number of stimuli differs.

Indeed, this is correct. The musical stimuli for the BAT were taken from the Beat Alignment Test of the Goldsmiths Musical Sophistication Index (Gold-MSI) v1.0 downloaded from https://www.gold.ac.uk/music-mind-brain/gold-msi/download/. This version was recommended to us by the authors of the 2014 study when we contacted them for stimuli, as Version 1.0 of the Gold-MSI is optimized relative to the version reported in Müllensiefen et al., 2014’s study, with 17 items, selected as described in the documentation available at https://www.gold.ac.uk/music-mind-brain/gold-msi/download/). We added a table outlining the BAT stimuli as supplementary material, and added a reference to the website and a specific version number of the BAT to clarify exactly which version was used. This also adds relevant information about the number of trials with phase and period shifts, as well as the amounts of those shifts on each trial.

L195-199: Musical stimuli were taken from the Beat Alignment Test of the Goldsmiths Musical Sophistication Index (Gold-MSI) v1.0 [26] downloaded from https://www.gold.ac.uk/music-mind-brain/gold-msi/download/. Version 1.0 of the Gold-MSI is optimized relative to the version reported in Müllensiefen et al., (2014), with 17 items, selected as described in the documentation available at https://www.gold.ac.uk/music-mind-brain/gold-msi/download/ (S1 Table).

L213-216: Phase shifts of the superimposed beeps were adjusted 10% or 17.5% ahead relative to the musical beat. Period shifts of the superimposed beeps were adjusted 2% slower or faster relative to the musical tempo (S1 Table).

- “low asynchrony scores reflected less variable and more consistent tap times” (l180). I do not think the asynchrony score reflect variability and consistency. It tells how far from the beat the participants tapped; a high score can reflect a very consistent performance in antiphase, for example. None of the two indexes reflect variability; it could be worth using a coefficient of variation of the asynchronies [SD(asynch)/Mean(asynch)], where asynch = Mean(Response - Beat). This index might reveal additional effects that the tempo matching index failed to capture. Notably, I expect that participants with dance and music training are less variable in their tapping performance. For example, a recent study on children with cerebellar anomalies showed that a dance training protocol reduced their variability in a synchronization task (Bégel et al., 2021).

This is a great suggestion. We have adjusted the interpretation for high/low asynchrony scores and analyzed coefficient of variability scores. The CoV data is now included in all results. 

Results

- “When assessing the effects of dance training on beat perception, the null model had a BF10 of 1 (Table 2b), suggesting that the data could not discriminate between the alternative hypothesis from the null hypothesis” (l288-290). This is also the case for the music & group model. Please explain a bit more how you come to this conclusion and what is the difference between the two Null models. This remark applies to the phase matching analyses as well.

We added a brief explanation of the 3 logical states (null hypothesis supported, null hypothesis rejected, insufficient information to discriminate between the two hypotheses) in relation to BF10 values in the “Bayesian analyses” paragraph. 

L391-401: The Bayesian ANOVA compares the predictive performance of each model with and without each independent variable and their interactions. The P(M) column is the prior model probability which assumes that all rival models are equally likely to represent the data. The P(M|data) column indicates the probability of each model given the actual data. The BFm column indicates the relative likelihood of each model compared to the average of all other models, and the BF10 column indicates the relative likelihood of each model compared to the null model. Generally, a BF10 < 1/10 provides strong support for the null hypothesis, BF10 > 10 provides strong support for the alternative model, and 1/3 < BF10 < 3 indicates that the data is insufficient to support either hypothesis [29,30]. The strength of the evidence can be quantified based on the Bayes Factor (e.g., BF10 =8 is twice as strong as BF10 = 4 in supporting the alternative hypothesis).

- Figure 2. Because the PD group was not included in the analyses with dance training, I think their results should not appear in the figure.

We removed the PD group from the dance figures.

- Figure 3. It is not clear what the three lines (user, wide, ultrawide) are. Most readers are not familiar with Bayesian statistics. It is important to give more information.

Thanks for noting this, we realized the sequential aspect of the analysis isn’t relevant to the analysis and have decided to remove the figure entirely. 

Discussion

- The absence of difference between the PD and healthy adults groups in beat perception is inconsistent with the results of Benoit et al. (2014). There are interindividual difference in Parkinson’s patients that may explain the contradictory results (Cochen de Cock et al., 2018). I think this should be mentioned in the discussion.

Thanks for the suggestion. We have referenced and discussed the Benoit et al. (2014) and Cochen et al., (2018) papers in our discussion section. 

L499-504: In contrast, our results differed from Benoit et al.’s findings that Parkinson’s patients showed worse timing perception than healthy adults [31]. However, in their study, Parkinson’s patients were tasked to detect misaligned beats in a two-measure music excerpt, as opposed to several seconds of tones overlaid on music in the BAT. This task is less taxing on memory and attention, and thus may not be able to differentiate between healthy adult groups and Parkinson’s patients.

L528-530: Cochen de Cock et al. corroborated these findings by suggesting that cognitive abilities such as attention, executive function and cognitive flexibility could influence beat perception abilities [32].

- The “Implications” and “Limitations” sections are quite long. Condensing them into one section seems appropriate.

We have condensed these paragraphs into one section. 

Reviewer #2

The main limitation is that the information about the training is very limited: we don’t know when the training was and what its intensity and level was. This an important limitation in this study that is discussed in the discussion.

We agree this is a limitation in our study and have noted that in our discussion section. 

L543-544: Years of training is an imprecise quantification of true training effects, as the rigor of different training programs and hours of deliberate practice varies across individuals. 

There is also an important lack of information on Parkinson’s disease. Hoehn and Yahr stage is very insufficient to evaluate PD severity. Disease duration Ldopa Equlivalent daily dose are needed to understand the kind of patients that were explored. These two information should be added.

Thanks for noting this limitation, we agree that the Hoehn and Yahr scale provides limited information on the PD severity. We unfortunately do not have other information on the PD participants regarding to their disease and have noted this limitation in our discussion section. 

L611-615: Hoehn and Yahr stage is also the only indicator of disease severity collected in the study, and future work addressing the impact of these factors across disease stages would benefit from including more specific disease severity information such as disease duration or the levodopa equivalent daily dose and by including participants with moderate to severe stages of Parkinson’s. 

Also the recent reference on rhtythm disturabances in PD and prePD should be added and discussed

The reference was added to our introduction section. 

L73-76: Furthermore, beat perception and production deficits are correlated with idiopathic REM sleep disorder which commonly occurs prior to Parkinson’s disease onset and is often considered a prodromal-Parkinson’s disease symptom [11].

---

## [Decision Letter · Decision Letter 1]

19 Jan 2022

PONE-D-21-33421R1The effects of Parkinson’s disease, music training, and dance training on beat perception and production abilitiesPLOS ONE

Dear Dr. Hsu,

Thank you for submitting your manuscript to PLOS ONE. After careful consideration, we feel that it has merit but does not fully meet PLOS ONE’s publication criteria as it currently stands. Therefore, we invite you to submit a revised version of the manuscript that addresses the points raised during the review process.

We look forward to receiving your revised manuscript.

Kind regards,

Sonja Kotz

Academic Editor

PLOS ONE

Journal Requirements:

Reviewers' comments:

Reviewer's Responses to Questions

**Comments to the Author**

1. If the authors have adequately addressed your comments raised in a previous round of review and you feel that this manuscript is now acceptable for publication, you may indicate that here to bypass the “Comments to the Author” section, enter your conflict of interest statement in the “Confidential to Editor” section, and submit your "Accept" recommendation.

Reviewer #1: (No Response)

Reviewer #2: All comments have been addressed

2. Is the manuscript technically sound, and do the data support the conclusions?

Reviewer #1: Yes

Reviewer #2: Yes

3. Has the statistical analysis been performed appropriately and rigorously? 

Reviewer #1: Yes

Reviewer #2: N/A

4. Have the authors made all data underlying the findings in their manuscript fully available?

Reviewer #1: Yes

Reviewer #2: Yes

5. Is the manuscript presented in an intelligible fashion and written in standard English?

Reviewer #1: Yes

Reviewer #2: Yes

6. Review Comments to the Author

Reviewer #1: Thank you for giving me the opportunity to review a new version of the manuscript. The authors carefully addressed my comments and I believe the article is much improved now. Nevertheless, I recommend further changes to the presentation of the Bayesian analyses results, which are still confusing. In my opinion, the article will be acceptable for publication in PlosOne after the authors modify this point. Please see the detail below. à

I believe there is still a problem with the sentence “When assessing the effects of dance training on beat perception, the null model had a BF10 of 1 (Table 2b), suggesting that the data could not discriminate between the alternative hypothesis from the null hypothesis”. Description of table 4 is similar. Again, all null models have a BF10 of 1, so how can that be supporting the null hypothesis in itself? I believe the reason is that all other models have a BF10 comprised between 1/3 and 3.

I suggest presenting the interpretation for the full range of possible BF10 values: “BF10 < 1/10 provides strong support for the null hypothesis, 1/10 < BF10 < 1/3 provides moderate support for the null hypothesis”, and again 3 < BF10 < 10. It would also be useful to explicitly say that the BF10 value is a ratio that is expressed in decimal in the tables. I had difficulties to understand that.

I am not convinced that scientific notation should be used in table 2. Numbers are not that big and scientific notation can be confusing. Three numbers presentations are used for BF10 when presenting the interpretation of the index and the values (ratios, decimals and scientific), which make it very hard to follow.

Reviewer #2: Thanks for your modifications, all the remarks have been adressed correctly, added in the results or as limitations

7. PLOS authors have the option to publish the peer review history of their article (what does this mean?). If published, this will include your full peer review and any attached files.

Reviewer #1: **Yes: **Valentin Bégel

Reviewer #2: No

---

## [Author Response · Author response to Decision Letter 1]

20 Jan 2022

Dear Dr. Sonja Kotz,

Thank you for handling our manuscript, “The effects of Parkinson’s disease, music training, and dance training on beat perception and production abilities,” for possible publication in Plos One. We also thank the reviewers for their careful feedback and helpful comments. We now submit a revised version, with further clarifications about our Bayesian analyses. Point-by-point responses to the reviewers’ comments are attached below. Note that the line numbers refer to the Revised Manuscript with Tracked Changes file. We hope that you might find this version to be suitable for publication for the wide readership of Plos One.

Sincerely,

Prisca Hsu, Emily Ready, Jessica Grahn

Reviewer #1

I believe there is still a problem with the sentence “When assessing the effects of dance training on beat perception, the null model had a BF10 of 1 (Table 2b), suggesting that the data could not discriminate between the alternative hypothesis from the null hypothesis”. Description of table 4 is similar. Again, all null models have a BF10 of 1, so how can that be supporting the null hypothesis in itself? I believe the reason is that all other models have a BF10 comprised between 1/3 and 3.

Thanks for pointing this out. We realize that the description of the BF10 may be confusing to our readers. As noted in L311-317, a BF10 value of 1 falls between 0.33 and 3 and thus indicates inconclusive results that do not clearly support H0 or H1 models. We have rephrased those sentences.

L331-333: When assessing the effects of dance training on beat perception, the null model had a BF10 of 1 (Table 2b), suggesting that the data are inconclusive and do not support either null or alternative models.

L367-369: For tempo matching, the Bayesian ANOVAs for both music and dance revealed a BF10 = 1 (Tables 4a & 4b). A BF10 = 1 value suggests that the data are inconclusive and do not support either null or alternative models.

I suggest presenting the interpretation for the full range of possible BF10 values: “BF10 < 1/10 provides strong support for the null hypothesis, 1/10 < BF10 < 1/3 provides moderate support for the null hypothesis”, and again 3 < BF10 < 10. It would also be useful to explicitly say that the BF10 value is a ratio that is expressed in decimal in the tables. I had difficulties to understand that.

This is great suggestion. We have adjusted the Bayesian statistics description and changed the BF10 values to decimal notation in the following lines.

L311-317: As BF10 deviates from 1, support for the null or alternative hypothesis increases. Generally, 0.33 < BF10 < 3 indicates that the data is insufficient to support either null or alternative hypothesis. 0.1 < BF10 < 0.33 provides moderate support for the null hypothesis and 3 < BF10 < 10 provides moderate support for the alternative hypothesis. Finally, a BF10 < 0.1 provides strong support for the null hypothesis and BF10 > 10 provides strong support for the alternative hypothesis. [29,30]. Strength of the evidence can be quantified based on the Bayes Factor (e.g., BF10 =8 is twice as strong as BF10 = 4 in supporting the alternative hypothesis).

I am not convinced that scientific notation should be used in table 2. Numbers are not that big and scientific notation can be confusing. Three numbers presentations are used for BF10 when presenting the interpretation of the index and the values (ratios, decimals and scientific), which make it very hard to follow.

Thanks for noting this. We have changed the notation of the numbers in table 2. We have also changed the BF10 values to decimal notation in the Bayesian Analyses section.

---

## [Decision Letter · Decision Letter 2]

25 Jan 2022

PONE-D-21-33421R2The effects of Parkinson’s disease, music training, and dance training on beat perception and production abilitiesPLOS ONE

Dear Dr. Hsu,

Thank you for submitting your manuscript to PLOS ONE. After careful consideration, we feel that it has merit but does not fully meet PLOS ONE’s publication criteria as it currently stands. Therefore, we invite you to submit a revised version of the manuscript that addresses the points raised during the review process.

We look forward to receiving your revised manuscript.

Kind regards,

Sonja Kotz

Academic Editor

PLOS ONE

Journal Requirements:

Reviewers' comments:

Reviewer's Responses to Questions

**Comments to the Author**

1. If the authors have adequately addressed your comments raised in a previous round of review and you feel that this manuscript is now acceptable for publication, you may indicate that here to bypass the “Comments to the Author” section, enter your conflict of interest statement in the “Confidential to Editor” section, and submit your "Accept" recommendation.

Reviewer #1: (No Response)

2. Is the manuscript technically sound, and do the data support the conclusions?

Reviewer #1: Yes

3. Has the statistical analysis been performed appropriately and rigorously? 

Reviewer #1: Yes

4. Have the authors made all data underlying the findings in their manuscript fully available?

Reviewer #1: Yes

5. Is the manuscript presented in an intelligible fashion and written in standard English?

Reviewer #1: Yes

6. Review Comments to the Author

Reviewer #1: Thanks a lot for addressing my comments; I think the presentation of the Bayesian analyses is much improved, which is very important as most readers are not familiar with it.

Nevertheless, I still think there is a major mistake with the sentence ‘When assessing the effects of dance training on beat perception, the null model had a BF10 of 1 (Table 2b), suggesting that the data are inconclusive’. I already pointed that in my previous reviews: all null models have a BF10 of 1, included the one presented in table 2a, where the authors say that data are conclusive. I believe that the null model’s BF is always 1. Unless I am missing something, this sentence is wrong and makes the Bayesian results really confusing. The sentence describing the model in the ‘Beat production: tempo matching’ section is also incorrect. Please modify that. I am sorry for delaying the decision on the article, but I do not think it can be published with such confusing mistakes.

7. PLOS authors have the option to publish the peer review history of their article (what does this mean?). If published, this will include your full peer review and any attached files.

Reviewer #1: **Yes: **Valentin Bégel

---

## [Author Response · Author response to Decision Letter 2]

28 Jan 2022

Dear Dr. Sonja Kotz,

Thank you for handling our manuscript, “The effects of Parkinson’s disease, music training, and dance training on beat perception and production abilities,” for possible publication in Plos One. We thank reviewer 1 for their careful feedback and helpful comments. We now submit a revised version, with further clarifications about our Bayesian analyses. We hope that you might find this version to be suitable for publication for the wide readership of Plos One.

Sincerely,

Prisca Hsu, Emily Ready, Jessica Grahn

Reviewer #1: 

Thanks a lot for addressing my comments; I think the presentation of the Bayesian analyses is much improved, which is very important as most readers are not familiar with it.

Nevertheless, I still think there is a major mistake with the sentence ‘When assessing the effects of dance training on beat perception, the null model had a BF10 of 1 (Table 2b), suggesting that the data are inconclusive’. I already pointed that in my previous reviews: all null models have a BF10 of 1, included the one presented in table 2a, where the authors say that data are conclusive. I believe that the null model’s BF is always 1. Unless I am missing something, this sentence is wrong and makes the Bayesian results really confusing. The sentence describing the model in the ‘Beat production: tempo matching’ section is also incorrect. Please modify that. 

Thank you for your careful review. You are correct. All BF10 values are relative to the BF10 value of the null model. As such, the BF10 value for the null model will always be 1. We realize that our indication of interpreting the BF10=1 is confusing and have modified those lines to be clearer about how the BF10 values should be interpreted. 

L325-327 When assessing the effects of dance training on beat perception, the dance model had a BF10 value of 0.33 (Table 2b), indicating insufficient support for either the null hypothesis or the alternative hypothesis [29,30]. 

L362-364 For tempo matching, the music model had a BF10 of 0.50 (Table 4a), indicating insufficient support for either the null hypothesis or alternative hypothesis. For dance, BF10 = 0.22 (Table 4b), indicating moderate support for the null hypothesis.

---

## [Decision Letter · Decision Letter 3]

14 Feb 2022

The effects of Parkinson’s disease, music training, and dance training on beat perception and production abilities

PONE-D-21-33421R3

Dear Dr. Hsu,

We’re pleased to inform you that your manuscript has been judged scientifically suitable for publication and will be formally accepted for publication once it meets all outstanding technical requirements.

Kind regards,

Sonja Kotz

Academic Editor

PLOS ONE

**Comments to the Author**

1. If the authors have adequately addressed your comments raised in a previous round of review and you feel that this manuscript is now acceptable for publication, you may indicate that here to bypass the “Comments to the Author” section, enter your conflict of interest statement in the “Confidential to Editor” section, and submit your "Accept" recommendation.

Reviewer #1: All comments have been addressed

2. Is the manuscript technically sound, and do the data support the conclusions?

Reviewer #1: Yes

3. Has the statistical analysis been performed appropriately and rigorously? 

Reviewer #1: Yes

4. Have the authors made all data underlying the findings in their manuscript fully available?

Reviewer #1: Yes

5. Is the manuscript presented in an intelligible fashion and written in standard English?

Reviewer #1: Yes

6. Review Comments to the Author

Reviewer #1: Thank you for addressing my final comments. It was important to clarify the presentation of the Bayesian analyses. In my view, the article is now suitable for publication and makes an important contribution to the field

7. PLOS authors have the option to publish the peer review history of their article (what does this mean?). If published, this will include your full peer review and any attached files.

Reviewer #1: **Yes: **Valentin Bégel

---

## [Editor Report · Acceptance letter]

23 Feb 2022

PONE-D-21-33421R3 

The effects of Parkinson’s disease, music training, and dance training on beat perception and production abilities 

Dear Dr. Hsu:

I'm pleased to inform you that your manuscript has been deemed suitable for publication in PLOS ONE. Congratulations! Your manuscript is now with our production department. 

Kind regards, 

on behalf of

Dr. Sonja Kotz 

Academic Editor

PLOS ONE